# Galectin-3: A Potential Prognostic and Diagnostic Marker for Heart Disease and Detection of Early Stage Pathology

**DOI:** 10.3390/biom10091277

**Published:** 2020-09-04

**Authors:** Akira Hara, Masayuki Niwa, Tomohiro Kanayama, Kei Noguchi, Ayumi Niwa, Mikiko Matsuo, Takahiro Kuroda, Yuichiro Hatano, Hideshi Okada, Hiroyuki Tomita

**Affiliations:** 1Department of Tumor Pathology, Gifu University Graduate School of Medicine, Gifu 501-1194, Japan; t_knym@gifu-u.ac.jp (T.K.); gucchan1013@gmail.com (K.N.); mulmirry@yahoo.co.jp (A.N.); kyokui100202@gmail.com (M.M.); longhong.heitian@gmail.com (T.K.); yuha@gifu-u.ac.jp (Y.H.); h_tomita@gifu-u.ac.jp (H.T.); 2Medical Education Development Center, Gifu University School of Medicine, Gifu 501-1194, Japan; mniwa@gifu-u.ac.jp; 3Department of Emergency and Disaster Medicine, Gifu University Graduate School of Medicine, Gifu 501-1194, Japan; hideshi@gifu-u.ac.jp

**Keywords:** galectin-3, biomarker, diagnostic, prognostic, early stage, heart disease, animal model

## Abstract

The use of molecular biomarkers for the early detection of heart disease, before their onset of symptoms, is an attractive novel approach. Ideal molecular biomarkers, those that are both sensitive and specific to heart disease, are likely to provide a much earlier diagnosis, thereby providing better treatment outcomes. Galectin-3 is expressed by various immune cells, including mast cells, histiocytes and macrophages, and plays an important role in diverse physiological functions. Since galectin-3 is readily expressed on the cell surface, and is readily secreted by injured and inflammatory cells, it has been suggested that cardiac galectin-3 could be a marker for cardiac disorders such as cardiac inflammation and fibrosis, depending on the specific pathogenesis. Thus, galectin-3 may be a novel candidate biomarker for the diagnosis, analysis and prognosis of various cardiac diseases, including heart failure. The goals of heart disease treatment are to prevent acute onset and to predict their occurrence by using the ideal molecular biomarkers. In this review, we discuss and summarize recent developments of galectin-3 as a next-generation molecular biomarker of heart disease. Furthermore, we describe how galectin-3 may be useful as a diagnostic marker for detecting the early stages of various heart diseases, which may contribute to improved early therapeutic interventions.

## 1. Introduction

Heart diseases are a leading cause of death worldwide, killing approximately 17.9 million people each year. Individuals at risk of heart disease may demonstrate an elevated body weight, blood pressure, plasma cholesterol and blood glucose, as well as obesity. These factors can be easily measured in primary healthcare services. In addition to these standard measures, the use of molecular biomarkers may provide a much earlier detection of heart disease, thereby providing earlier and more efficacious therapeutic interventions. The detection of ideal molecular biomarkers, those that are both sensitive and specific to heart disease, are likely to provide an early diagnosis and suggest specific targeted therapy. However, to date, such ideal biomarkers of heart disease have yet to be identified, despite advances in technologies such as multiplex molecular and genetic biomarkers [1,2]. Thus, the aim of this review is to provide an overview of a candidate molecular heart disease biomarker, galectin-3 (Gal-3).

Galectins are composed of a family of widely expressed β-galactoside-binding lectins and can modulate basic cellular functions such as “cell-to-cell” and “cell-to-matrix” interactions, cell growth and differentiation, tissue regeneration and the regulation of immune cell activities [3,4,5]. Galectins have been classified according to their carbohydrate recognition domain (CRD) number and function. The CRDs recognize β-galactoside residues that form complexes that crosslink glycosylated ligands [6,7,8]. The following three types of galectin members are widely accepted (Figure 1): (1) prototype galectins (galectin-1, -2, -5, -7, -10, -11, -13, -14, and -15), containing a single CRD that form noncovalent homodimers; (2) tandem-repeat galectins (galectin-4, -6, -8, -9, and -12), carrying two CRD motifs connected by a peptide linker and (3) a chimera-type galectin (Gal-3), which is characterized by having a single CRD and an amino-terminal polypeptide tail region [4,7,8]. The members of galectins, numbered consecutively by order of discovery, are ubiquitously present in vertebrates, invertebrates and, also, protists [3].

One member of the family, Gal-3, an approximately 30-kDa chimera-type galectin, is expressed by various immune cells, including mast cells, histiocytes and macrophages, which are associated with the mononuclear phagocytic system in various tissues [9]. Although Gal-3 is predominantly present as a cytosolic protein for cellular function and a nuclei protein for splicing, it is also expressed on cell surfaces and secreted into the plasma by various cells [10]. It has been shown that Gal-3 plays an important role in diverse physiological functions, such as cell growth and differentiation, macrophage activation, angiogenesis, apoptosis and antimicrobial activity, as well as acting as a mediator of local inflammatory responses in many pathological conditions [11].

Since Gal-3 is readily expressed on the cell surface, and easily secreted into biological fluids (e.g., serum and urine) from injured cells and inflammatory cells, recent studies suggest that cardiac Gal-3 could be a marker for cardiac disorders such as cardiac inflammation and fibrosis, depending on the specific pathogenesis of human heart diseases [12,13]. Therefore, Gal-3 may be a novel candidate biomarker for the diagnosis, analysis and prognosis of various cardiac diseases, including heart failure [14,15,16,17].

Furthermore, Gal-3 may also be useful for detecting the early stages of some diseases. Gal-3 has already been used as a possible clinical biomarker in the early detection of myocardial dysfunction, including acute heart failure [17]. In experimental acute myocarditis following viral infection, Gal-3 has been validated as a biomarker of cardiac fibrotic degeneration in animal models [13,16]. Serum Gal-3 levels have been used as an early diagnostic biomarker for detecting cardiac degeneration in acute myocarditis [13] and acute myocardial infarction [16].

Established cardiovascular biomarkers, other than Gal-3, have been investigated for many years for their ability to differentiate different pathophysiological processes, such as inflammation, injury and fibrosis. These biomarkers have been used in clinical practices to reveal the pathophysiological characteristics of heart failure, myocyte injury, ventricular wall stress, fibrosis and cardiac remodeling. Natriuretic peptides (NPs), soluble ST2 (suppression of tumourigenicity2) (sST2), myocardial troponin I (cTnI), myocardial troponin T (cTnT), C-reactive protein (CRP) and growth and differentiation factor-15 (GDF-15) are the cardiovascular biomarkers discussed in this review.

In this review, we discuss and summarize the recent developments of Gal-3 as a next-generation molecular biomarker in not only the patients with various types of heart diseases but, also, the disease-associated animal models. Furthermore, we provide a possibility of Gal-3 as a diagnostic or prognostic marker for detecting the early stages of various heart diseases.

## 2. Current Clinical Studies of Gal-3 as a Possible Biomarker in Heart Disease

Clinically, Gal-3 is studied most intensively in heart disease as a diagnostic or prognostic marker [14,15,16,17]. In addition to heart disease, Gal-3 has also been considered as a biomarker in other human diseases, such as viral infections [18,19,20], autoimmune diseases [21,22,23,24], diabetes [25,26,27], kidney disease [25,26,28,29] and even tumor formations, including thyroid tumors [30,31,32,33,34,35,36,37,38,39]. The diverse clinical involvement of galectins in many diseases has been suggested as a role for the regulators of acute and chronic inflammation, which is linking inflammation-related macrophages to the promotion of fibrosis [40]. The evidence suggests that Gal-3 is not an organ-specific marker but a specific marker of individual pathogenesis, such as inflammation or fibrosis. Therefore, the primary sources for circulating Gal-3 are not always identified.

Many clinical studies of heart failure suggest that plasma and cardiac Gal-3 levels reflect cardiac inflammatory responses and can be considered as a possible marker for both cardiac inflammation and fibrosis, depending on the pathogenesis of heart failure [40]. However, the mechanism responsible for increased blood levels of Gal-3 remains incompletely defined. Several studies have been conducted on Gal-3 to assess its prognostic effect in heart failure populations. In general, a high concentration of plasma Gal-3 correlates with a clinical outcome in heart failure associated with cardiac fibrosis [41,42]. The increased plasma levels of Gal-3 are associated with adverse long-term cardiovascular outcomes in both patients with acute [43,44] and chronic [45,46] heart failure. However, some studies have generated conflicting results and suggested that Gal-3 is a poor predictor of mortality [47]. In addition, some studies have reported contradictory results on the association between plasma and cardiac Gal-3 levels and cardiac fibrosis in heart failure [48,49,50]. These clinical studies were limited by their small sample sizes and nondetailed evaluations. However, a large-scale meta-analysis of the plasma Gal-3 in the general population has revealed that elevated plasma galectin-3 is associated with a high risk of cardiovascular mortality and heart failure, in addition to all-cause mortality, and has suggested that galectin-3 is an important prognostic factor for patients with heart disease [51].

Various heart diseases, such as myocardial infarction, myocarditis, hypertension and subsequent heart failure, have dynamic interactions between inflammation and fibrosis [52]. Furthermore, recent studies indicate that Gal-3 is involved in cardiovascular fibrosis as a regulatory molecule in heart failure and, thus, that Gal-3 inhibition ameliorates myocardial injury, highlighting its therapeutic potential [53,54]. Atrial fibrillation, the most common arrhythmia presented in clinical practice, can occur in association with electrical and structural remodeling in the atria. Several lines of evidence demonstrate that myocardial strain, fibrosis and inflammation are involved in the pathogenesis of arrhythmia, including atrial fibrillation, in addition to conventional factors such as the increased left atrial size and the presence of heart failure, coronary heart disease or valvular heart disease. Galectin-3 may be involved in atrial structural remodeling, which involves progressive fibrogenesis in atrial fibrillation patients [55]. A meta-analysis of the relationship between baseline circulating Gal-3 levels and the recurrence of atrial fibrillation in patients undergoing catheter ablation showed that baseline circulating Gal-3 levels were significantly higher in patients with a recurrence of atrial fibrillation compared to those without atrial fibrillation [56]. In addition, higher baseline Gal-3 levels were independently associated with a significantly higher risk of recurrence of atrial fibrillation after catheter ablation [56].

Gal-3 is also reported to be elevated in patients with adult congenital heart disease. A significant association of Gal-3 with functional capacity, cardiac function and adverse cardiovascular events in patients with adult congenital heart disease has been reported recently [57]. In pediatric heart surgery, elevated pre-and postoperative levels of Gal-3 are reported to be associated with an increased risk of readmission or mortality after the operation [58]. Thus, the clinically available biomarker Gal-3 can be used for improved risk stratification.

Chronic kidney disease (CKD) is a risk factor for cardiovascular disease (CVD). Many cardiac biomarkers associated with heart diseases may also reflect the progression of kidney disease. It is plausible that CKD and CVD are closely interrelated, and patients with CKD have a strong risk of CVD [59,60]. Gal-3 is associated with myofibroblast proliferation, fibrogenesis, tissue repair and myocardial remodeling and is also associated with kidney fibrosis and an increased risk of CKD. Thus, the wide tissue distribution of Gal-3 associated with fibrosis in both CVD and CKD complicates the utility of Gal-3 as a cardiac biomarker in CKD patients [28]. Furthermore, a strong and negative correlation between circulating Gal-3 levels and the estimated glomerular filtration rate has been reported. Renal dysfunction is a determinant of blood Gal-3 levels, and the Gal-3 levels are markedly elevated in patients with severe renal failure [61,62,63]. This means that high concentrations of Gal-3 may be associated with the progression of CKD [26]. Furthermore, Gal-3 is reported to play a pivotal role in renal interstitial fibrosis and the progression of CKD [64]. A glomerular Gal-3 expression was observed in 81.8% of patients with systemic lupus erythematosus (SLE) nephritis but not in the control patients [65]. Blood Gal- 3 levels were particularly higher in SLE patients with nephritis than in healthy controls. Gal-3 may contribute to the glomerulonephritis in SLE, and thus, the inhibition of Gal-3 may be a promising therapeutic strategy to prevent advanced renal disease.

The potential use of Gal-3 as a diagnostic biomarker and prognostic indicator in various heart diseases is summarized in Table 1.

## 3. Current Guidelines for the Clinical Use of Biomarkers in Heart Disease

The clinical use of established or recommended biomarkers in the diagnosis and risk management of heart failure has been indicated by some representative guidelines. The Heart Failure Society of America (HFSA), European Society of Cardiology (ESC) and American College of Cardiology Foundation (ACC)/American Heart Association (AHA) have indicated that the NPs, circulating hormones secreted by cardiomyocytes in the heart ventricles, play an important role in the regulation of the intravascular blood volume and vascular tone and act as useful diagnostic biomarkers in patients suspected of heart failure [46,68,69,70]. Guideline management based on biomarkers has brought a new dimension in diagnosis, prognosis and treatment evaluation. However, the utilities of novel biomarkers other than NPs are not well-established in clinical routine analyses. The National Academy of Clinical Biochemistry (NACB) recommended the clinical assessment and analytical perspectives of novel biomarkers in the diagnosis and management of heart failure [71]. The novel biomarkers in these criteria need to be able to recognize the fundamental causes of heart failure, assess their severity and foresee the risk of disease progression. In fact, with regards to Gal-3 as a novel biomarker, the ACC/AHA guidelines recommended the use of Gal-3 for the assessment of cardiac fibrosis in heart failure; however, thus far, the ESC has not recommended the clinical use of Gal-3 [72].

## 4. Established Cardiovascular Biomarkers Other than Gal-3

As mentioned in the above section, beside the recommendation of NPs by several guidelines on heart failure, many other biomarkers have been investigated as to whether they could reflect different pathophysiological processes such as inflammation, injury and fibrosis. In fact, many candidate protein markers reveal the pathophysiological characteristics of heart failure, including inflammation, myocyte injury, biochemical wall stress, fibrosis and cardiac remodeling. Below, we describe established and novel biomarkers for heart disease.

### 4.1. NPs

Since the first discovery of NP structures and functions in humans in 1984, three types of NPs have been identified in mammals: atrial natriuretic diuretic peptide (ANP), cerebral natriuretic peptide (BNP) and C-type natriuretic peptide (CNP). In particular, BNP and N-terminal-proBNP (the prohormone proBNP is cleaved to the active BNP and the inactive amino acid N-terminal proBNP (NT-proBNP)) are the gold standard clinical diagnostic biomarkers as heart failure biomarkers [73]. In healthy adults, BNP blood levels are less than 25 pg/mL, and NT-proBNP levels are less than 70 pg/mL [74].

Heart failure is a complex, progressive clinical condition in which the heart fails to pump enough blood to supply the body with the amount of blood it needs. Heart failure is a progressive condition that is accompanied by sudden dysfunction. The rapid and accurate diagnosis of heart failure is essential when the progression of the disease is rapid. The diagnosis of heart failure is based on a physical examination and the patient’s history, and additional diagnostic tests such as electrocardiography, chest radiography, echocardiography and NT-proBNP have been found to be useful as a means of the further detailed diagnosis of heart failure.

According to the 2016 ESC guidelines [69], measuring plasma NPs can help differentiate both nonacute and acute heart from noncardiac conditions. However, high levels of NPs do not definitively confirm heart failure; therefore, the use of NPs is not recommended to establish the final diagnosis.

It is recommended to use plasma NP concentrations as a clinical test at the first visit of patients with nonacute symptoms if echocardiography is not rapidly available: NT- proBNP < 125 pg/ml = a low probability of heart failure.

A similar concept in the case of acute symptoms but with a higher cut-off value: NT-proBNP < 300 pg/mL = less chance of heart failure. The guideline recommends differentiating acute heart failure from acute dyspnea of noncardiac origin by measuring NT-proBNP in emergency patients with suspected acute dyspnea or acute heart failure.

It is widely recognized that the mechanisms that contribute to the development of heart failure include a complex bidirectional interaction between the kidney and the heart, which is expressed in the term cardiac-renal syndrome (CRS). In a wave of new urinary biomarkers associated with CRS, CNP has emerged as an innovative biomarker of renal structural and functional impairment in heart failure and chronic renal disease states. CNP as a diagnostic and prognostic biomarker in heart failure and renal disease states is expected to have future clinical utility [75].

### 4.2. Soluble ST2 (sST2)

ST2 (suppression of tumourigenicity2) is a member of the interleukin (IL)-1 family and has both a membrane-bound receptor type (ST2L) and soluble (sST2) isoform. In the physiological stretch state of the heart, myofibroblasts release IL-33, which binds to ST2L and promotes cell survival and integrity. This ST2L/IL-33 signaling is regulated by the sST2, which is a decoy of IL-33 secreted by cardiac fibroblasts in response to cardiac pressure and volume overload [76]. However, when local and neighboring cells abnormally increase the release of sST2, it excessively blocks IL-33/ST2L-binding, which is detrimental to the heart. That is, sST2 acts as a decoy receptor for IL-33 to regulate excessive IL33 signaling under normal conditions, but under pathological conditions, it excessively represses IL-33 signaling, resulting in the interruption of ST2L-mediated cardioprotection. This imbalance in sST2 levels in the extracellular space of the heart is strongly associated with major cardiovascular disorders, including coronary artery disease, heart failure and valvular heart disease [77,78]. Thus, sST2 has come to be used as a biomarker of cardiac stress and fibrosis, and its circulating blood levels are now approved as an additional stratification factor for heart failure [79] and as a biomarker of ventricular remodeling and fibrosis, along with Gal-3 [46].

Recent studies have demonstrated that elevated ST2 levels in acute heart failure are prognostic for both recurrent hospitalization and mortality [80] and that ST2 levels in response to drug treatments are associated with improved outcomes in patients with chronic heart failure [81]. Thus, while sST2 is a biomarker of myocardial wall stress and the activation of the fibrosis pathway, sST2 is also expressed in organs other than the heart and is not specific to heart failure, making its use for diagnostic purposes in non-heart failure patients problematic.

### 4.3. Myocardial Troponin I (cTnI) and Myocardial Troponin T (cTnT)

The troponin complex, consisting of three subunits: Troponin T, I and C, regulates calcium-mediated muscle contractions between actin and myosin in both skeletal and cardiac muscles. The cardiac-specific isoforms of the troponin subunits cTnI and cTnT have very low or barely detectable blood levels in normal myocardium, but the blood levels of cTnI and cTnT are elevated when myocardial infarction damages cardiomyocytes. They are currently considered to be the most specific markers of myocardial damage, and clinical tests of cTnI and cTnT have been found to be clinically useful for the relative mortality risk classification of patients with acute coronary syndrome (ACS). The system for measuring cardiac troponin in the blood uses cardiac-specific antibodies that do not cross-react with skeletal muscle. Cardiac troponin is the diagnostic criteria for acute myocardial infarction [82,83].

### 4.4. C-reactive Protein (CRP)

CRP is a nonspecific blood marker of biological disease. The measurement of plasma CRP levels has proven clinically useful in the diagnosis and management of infectious diseases and the monitoring of a variety of noninfectious inflammatory diseases, including heart disease.

The importance of high-sensitive C-reactive protein (hs-CRP) measurements has also been reported. One small cohort study concluded that about 70% of patients with hs-CRP values above 4.25 mg/L at 90-day hospitalization died, compared to only 6.5% of patients with hs-CRP values below 4.25 mg/L [84]. Of note, Japanese people are characterized by lower mean CRP levels (one-third to one-fourth) compared to Westerners; however, a large cohort study revealed that higher levels of hs-CRP are associated with an increased risk of cardiovascular death and myocardial infarction, which may be useful in assessing cardiovascular disease risk [85,86].

### 4.5. Growth and Differentiation Factor-15 (GDF-15)

GDF-15, a member of the transforming growth factor-beta superfamily, also known as macrophage inhibitory cytokine-1 (MIC-1) or nonsteroidal anti-inflammatory drug-activating gene (NAG-1), has been implicated in pathologies such as inflammation, cancer, cardiovascular disease, lung disease and kidney disease. Cardiomyocytes produce and secrete GDF-15 in response to oxidative stress, stimulation by angiotensin II or proinflammatory cytokines, ischemia and mechanical stretch. Cell sources other than cardiomyocytes are known to include macrophages, vascular smooth muscle cells, endothelial cells and adipocytes, which secrete GDF-15 in response to oxidative or metabolic stress or stimulation by proinflammatory cytokines. GDF-15 is thought to protect the heart and adipose tissue, as well as endothelial cells, by inhibiting JNK (c-Jun N-terminal kinase), Bad (Bcl-2-associated death promoter) and EGFR (epidermal growth factor receptor) and activating the Smad, eNOS, PI3K and AKT signaling pathways [87].

GDF-15 can be used as a prognostic marker in patients with cardiovascular disorders in combination with conventional prognostic factors such as NT-proBNP and hs-TnT, as it is induced in hypertrophic and dilated cardiomyopathy after volume overload, ischemia and heart failure [88]. GDF-15 has also been shown to predict both the morbidity and mortality of CVD and cancer in apparently healthy older men [89]. It is interesting to suggest here that GDF-15 expression may be a common early indicator of cellular vulnerability to the development of vascular and cancer diseases. Measurements of sST2, hs-TnI and GDF-15 in the general population have also shown that sST2, GDF-15 and hs-TnI, in addition to established biomarkers such as hs-CRP, can predict cardiovascular risks [90]. GDF-15 has also been widely studied for its usefulness as a biomarker of cardiovascular events in diabetic patients, and it is interesting to note that GDF-15 was the only biomarker associated with cardiovascular events in patients with type 2 diabetes [91]. It has also been suggested that GDF-15 may be a new biomarker for identifying high-risk patients with muscle wasting and kidney dysfunction prior to cardiovascular surgery [92].

In a recent study of three biomarkers: Galectin 3, sSt2 and GDF-15 in adult CKD patients, higher circulating concentrations of all of them were associated with higher mortality, but only elevated GDF-15 concentrations were associated with an increased incidence of heart failure [93].

Finally, many biomarkers for heart disease, including Gal-3, have low tissue specificity, so it will be necessary to study them in combination as multiple markers rather than using them alone.

## 5. Gal-3 as a Biomarker of Cardiac Fibrosis

Cardiac inflammation and fibrosis are tightly implicated in the pathophysiological mechanisms for the myocardial tissue remodeling of heart failure regardless of its etiology [52]. As the important cellular and molecular mechanisms contributing to heart failure, the US Food and Drug Administration has approved Gal-3 as a soluble biomarker for cardiac fibrosis to detect cardiac tissue remodeling [94]. Thus, the serum levels of Gal-3 are associated with cardiac tissue remodeling and cardiac function. However, whether and how Gal-3 contributes to pathophysiology in cardiac remodeling remains unclear, especially in clinical settings. Although certain biomarkers involved in extracellular matrix turnover such as matrix metalloproteinase-3 and monocyte chemoattractant protein-1 at baseline were highly associated with the pathophysiology of acute myocardial infarction, the serum levels of Gal-3 were not related to the left ventricular remodeling defined by cardiac MRI in patients showing cardiac dysfunction [67].

The diverse clinical involvement of galectins in many diseases suggests its role as a regulator of acute and chronic inflammation, linking inflammation-related macrophages to the promotion of fibrosis [52,95]. Specifically, Gal-3 expression and secretion by macrophages is a major mechanism linking macrophages to fibrosis. Macrophages are increasingly recognized as a potential therapeutic target in cardiac fibrosis through interactions with connective tissue fibroblasts [96].

## 6. Usefulness of Gal-3 in Animal Models

The use of animal models that reproduce the clinical features of heart failure and heart disease have contributed to new approaches to improve diagnostic techniques and preventive/therapeutic strategies. As mentioned above, the roles of Gal-3 in heart failure and heart disease in humans are still controversial; however, many animal models have greatly improved our understanding of Gal-3 as a novel biomarker of heart disease. On the other hand, a few studies in animal models have generated conflicting results and suggested that Gal-3 is not a critical disease mediator of cardiac disease [97,98].

The overexpression of cardiac Gal-3 during early pre-symptomatic stages has been demonstrated to induce heart failure and heart disease in several studies using animal models. The intrapericardial injection of recombinant Gal-3 in healthy rats significantly increased the prevalence of cardiac fibrosis with cardiac remodeling and dysfunction and the induction of heart failure [99,100]. Gal-3 was also found to be colocalized with cardiac-infiltrating macrophages [99]. In contrast, cardiac remodeling and dysfunction induced by Gal-3 was prevented by a pharmacological inhibitor of Gal-3, N-acetyl-seryl-aspartyl-lysyl-proline [99]. An early increase in Gal-3 expression occurs in hypertrophied hearts, prior to the development of heart failure in a rat model of heart failure, with Gal-3 inducing cardiac fibroblast proliferation, collagen deposition and ventricular dysfunction [100]. This suggests that Gal-3 may be a novel biomarker candidate for the early stages of heart failure and that antagonizing Gal-3 at the early stages of heart failure may be a useful novel heart failure therapy. In a rat model subjected to pulmonary artery banding to induce right ventricular heart failure, Gal-3 was significantly increased in both the right and left ventricles, and protein kinase C promoted cardiac fibrosis and heart failure by stimulating the Gal-3 expression [101].

A myocardial ischemia/reperfusion (IR) injury is caused by reperfusion to restore the coronary blood flow to the ischemic region. IR also initiates an inflammatory response, contributing to adverse ventricular remodeling, which is possibly promoted by Gal-3. The upregulation of Gal-3, contributing to IR-induced cardiac dysfunction in a mouse model, has been reported [53]. Gal-3 inhibition ameliorates myocardial injury and suggests its therapeutic potential. In a rat model of IR injury induced by coronary artery ligation, a Gal-3 blockade improved ischemic injury through lower myocardial inflammation and reduced fibrosis [102]. In a mouse model of IR injury in the heart using wild-type and Gal-3 knockout mice, Gal-3 was shown to influence the redox pathways, control cell survival and death and play a protective role in the myocardium following IR injury [103].

In order to clarify the important role of cardiac Gal-3 expression during the early stage of heart failure, the time-course analysis of cardiac and serum Gal-3 in viral myocarditis, which was induced at 12, 24, 48, 96, 168 and 240 hours after a specific virus inoculation, was performed using a mouse model [13]. Gal-3 was demonstrated as a useful histological biomarker of cardiac fibrosis in acute myocarditis following a viral infection, and serum Gal-3 levels could be used as an early diagnostic marker for detecting cardiac fibrotic degeneration in acute myocarditis [13].

As mentioned earlier, Gal-3 expression and secretion by macrophages is a major function of macrophages not only contributing to excessive macrophage accumulation and their activation in cardiac tissue but, also, promoting fibroblast activation and proliferation, thus leading to cardiac fibrosis and cardiac remodeling [96,104]. In a mouse model of coxsackievirus B3 (CVB3)-induced myocarditis, mice infected with CVB3 and depleted of macrophages by a liposome-encapsulated clodronate treatment presented a reduction of acute myocarditis and chronic fibrosis, compared with untreated CVB3-infected mice [105]. In a pressure-overloaded mouse model of heart failure, Gal-3 interacted with aldosterone in promoting macrophage infiltration and cardiac fibrosis. The pharmacological inhibition of Gal-3 prevented the expression of genes involved in fibrogenesis (collagen type 1 and collagen type 3) and macrophage infiltration and cardiac remodeling [106]. Interestingly, in a pressure-overloaded mouse model, induced by transverse aortic constriction, an early upregulation of Gal-3 occurred three days after transverse aortic constriction in subpopulations of macrophages showing interstitial infiltration [97]. In contrast, large amounts of Gal-3 were localized in a subset of cardiomyocytes adjacent to fibrotic areas after 7–28 days of transverse aortic constriction [97]. The results indicate that the Gal-3 expressing cells change depending on the stage (early to late) of disease. Furthermore, these results from animal models indicate that cardiac-infiltrating macrophages expressing Gal-3 in the early stage are potential therapeutic targets for cardiac fibrosis and remodeling. Therefore, the early detection of such Gal-3-producing macrophages by a diagnostic marker is important.

Gal-3 is a key modulator of macrophages for differentiation or activation [107]. In a mouse model for acute myocardial infarction, the treatment of intravenous transplantation using human umbilical cord blood mesenchymal stem cells by modulating the conversion of macrophage subtype M1/M2 reduced the inflammatory response, decreased the serum Gal-3 level, improved cardiac function and protected the infarcted myocardium [108]. The serum level of Gal-3 is closely associated with the ratio of M1 macrophages to M2 macrophages, which is an important factor to improve cardiac function and protect the infarcted myocardium [108].

Representative microphotographs in cardiac lesions showing clear Gal-3 expression are demonstrated in Figure 2. The cardiac lesions of dilated cardiomyopathy in the late stage of δ-sarcoglycan (SG) knockout (KO) mice [13] is shown. The cardiac fibrous lesions, including tissue-resident macrophages, which are usually called histiocytes as a histomorphological term, are seen, with fibroblasts and collagen detected as blue in azan staining. Many histiocytes in the lesions are clearly seen as dark brown in Gal-3 immunostaining.

The promising animal models reproducing the clinical features of Gal-3 in heart failure and heart disease are summarized in Table 2.

## 7. Clinical Use of Gal-3 as a Next-generation Biomarker in the Future

As mentioned earlier, the clinical data has not shown that circulating Gal-3 levels reflect cardiac Gal-3 levels or cardiac fibrosis, although circulating Gal-3 has been demonstrated as a potential predictor for clinical outcomes in several cohort studies [41,42].

In a clinical setting, since various degrees of cardiac inflammation and the progression of fibrosis may be present in a patient with heart disease, blood Gal-3 levels may reflect a sum of different stages of pathophysiological conditions [12]. This is because the circulating blood levels of Gal-3 in a patient with various stages of heart disease cannot adequately reflect cardiac inflammation and fibrosis.

An endomyocardial biopsy is widely used as a diagnostic tool for patients with heart disease, such as myocarditis and secondary cardiomyopathies, which are often difficult to diagnose by conventional imaging alone [109]. There are many variables in human biopsy material by its nature, unlike those obtained from experimental animals. Human biopsies are usually performed under different conditions, variable time periods between biopsy and processing and variations in disease onset or severity. However, the histological examination of an endomyocardial biopsy is still the gold standard for the final diagnosis, despite continued advancements in diagnostic and therapeutic strategies [110,111,112].

In contrast to the clinical data, the blood levels of Gal-3 reflect the cardiac Gal-3 expression or cardiac fibrosis by using a sophisticated animal model for the time-course histological examination. Especially in the early phase of pathophysiology, there is a close relationship between the infiltration of Gal-3-positive macrophages and fibrotic lesions following myocarditis, and the blood levels of Gal-3 are tightly correlated with the number of cardiac Gal-3-positive cells [13]. The difference between the experimental data from animal studies and clinical findings from individual patients is due to a wide variability in clinical settings, with differences in sample collections and disease stages or severity.

Since experimental data from animal studies clearly indicate that the blood level of Gal-3 might be an early diagnostic biomarker for cardiac degeneration or fibrosis in acute myocarditis [13], further studies are needed to investigate whether such findings are also observed in cardiac degeneration or fibrosis in human patients. Gal-3 can be used reliably as a predictive biomarker for the early stage or new onset of heart disease, especially if it is derived from only the first single pathological lesion, without complicated factors. In addition, Gal-3 can also possibly be used in late stages of the diseases as an additional indicator for detecting a worse prognosis, mortality and readmission.

## 8. Conclusions and Perspectives

The blood levels of Gal-3 are altered by different clinical factors depending on the underlying pathophysiological conditions in patients, and thus, Gal-3 itself is not an organ-specific marker. However, Gal-3 is a specific marker of pathogenesis, such as macrophage-related disease or fibrosis, and the cardiac-infiltrating macrophages expressing Gal-3 in the early stages are potential therapeutic targets for cardiac fibrosis and remodeling. Therefore, the early detection of such Gal-3-producing macrophages by a diagnostic marker is important. Furthermore, Gal-3 is being tested for personalized medicine based on biomarker-guided diagnostics, using new technologies such as genetic biomarkers and multiplex biomarkers, combining multiple markers into a multiplex panel. In pediatric heart surgery, the clinically available biomarker Gal-3 can be used for improved risk stratification, because Gal-3 has recently been reported to be associated with an increased risk of readmission or mortality after the operation. In addition, Gal-3 at the early stages of inflammatory responses may be a potential therapeutic target for diseases, especially in cardiac fibrosis, autoimmune diseases, neurodegenerative diseases and cardio- and cerebrovascular diseases.

## Figures and Tables

**Figure 1 biomolecules-10-01277-f001:**
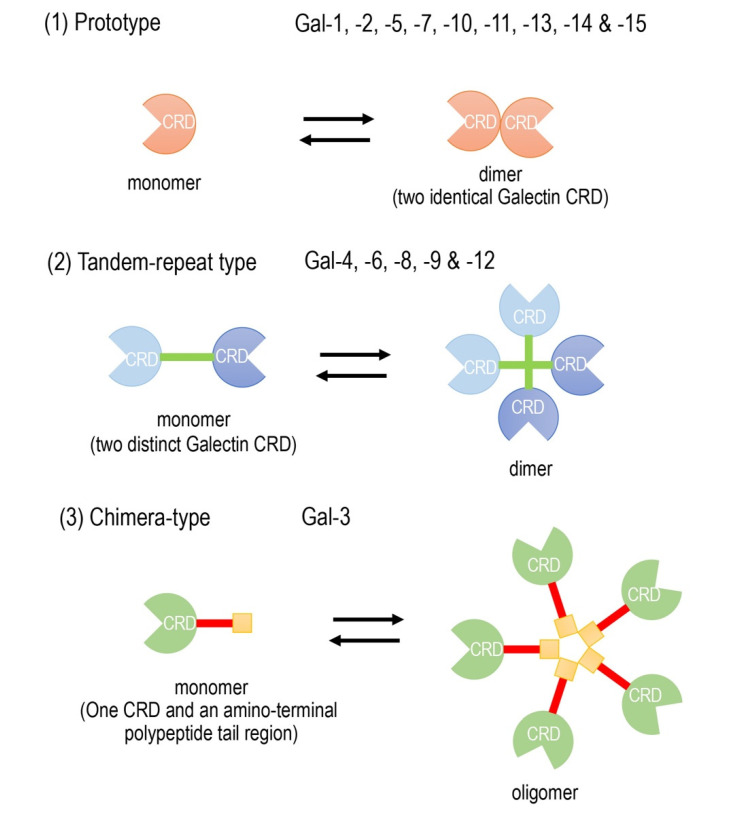
Schematic diagram of the galectin family members. Galectin members are divided into three types based on the organization of the galectin carbohydrate recognition domain (CRD).

**Figure 2 biomolecules-10-01277-f002:**
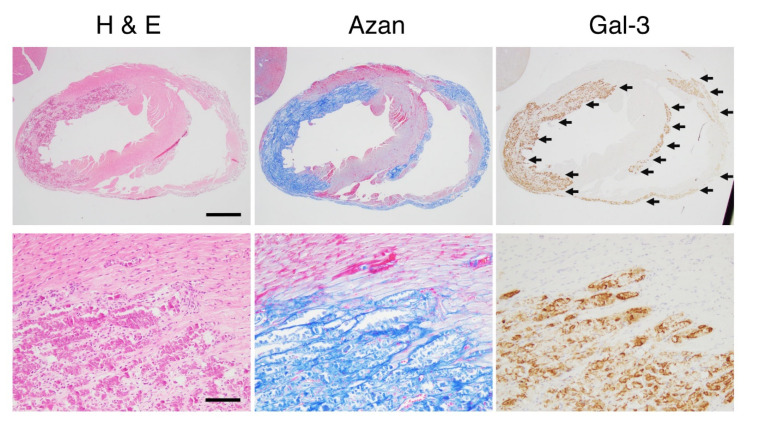
The cardiac lesions of dilated cardiomyopathy in the late stage of δ-sarcoglycan (δ-SG) knockout (KO) mice. Microphotographs for hematoxylin and eosin (H&E) staining, Azan staining and immunohistochemistry of Gal-3 are shown. Scale bars in H&E = 1 mm in the upper panel and 100 μm in the lower panel. Gal-3 expression sites indicated by arrows are identical to the fibrotic areas detected as blue in azan staining.

**Table 1 biomolecules-10-01277-t001:** The potential use of Gal-3 as a diagnostic biomarker and prognostic indicator in various heart diseases.

	Heart Disease	Usage of Biomarker	Potential Use as Biomarkers	Refs.
**Diagnostic Biomarkers**	acute heart failure	plasma level	• combination with natriuretic peptide	[43]
	acute heart failure	plasma level	• promising prognostic marker	[44]
	chronic heart failure	plasma level	• useful in heart failure	[66]
	chronic heart failure	myocardial and plasma level	• no association with histology	[45]
	acute myocardial infarction	serum level	• no definite relationship with ventricular remodeling	[67]
	chronic heart failure	myocardial and plasma level	• marker for both cardiac inflammation and fibrosis • circulating Gal-3 do not reflect cardiac fibrosis	[12]
**Prognostic Indicators**	chronic heart failure	plasma level	• association of Gal-3 with increased risk for incident heart failure and mortality	[41]
	cardiovascular disease	plasma level	• association of Gal-3 with age and risk factors of cardiovascular disease	[42]
	chronic heart failure	plasma level	• not suggested to be a predictor of mortality • candidate marker of a multi-biomarker panel in prognostication	[47]
	chronic heart failure	plasma level	• association of Gal-3 with severe heart failure • no prediction of outcomes after device implantation	[48]
	heart failure undergoing heart transplantation	plasma levelmyocardial Gal-3 expression	• insufficient use of Gal-3 as a marker of heart • local expression of myocardial Gal-3	[49]
	heart failure of hypertensive origin	biopsies and plasma samples	• cardiac and systemic excess Gal-3 in heart failure patients • no association with histology	[50]
	cardiovascular mortality and heart failure	plasma level	• large-scale meta-analysis • important prognostic value for heart disease	[51]
	atrial fibrillation	circulating Gal-3 level	• significantly higher in patients with recurrence of atrial fibrillation	[56]
	adult congenital heart disease	serum level	• association of Gal-3 with adverse cardiovascular events	[57]
	pediatric congenital heart disease	serum level	• association of Gal-3 with increased risk of readmission or mortality after the operation	[58]

**Table 2 biomolecules-10-01277-t002:** Promising animal models reproducing the clinical features of Gal-3 in heart failure and cardiovascular disease. IR: ischemia/reperfusion.

Animal Species	Experimental Models	Experimental Methods	Experimental Findings	Refs.
rat	chronic heart failure	intrapericardial injection of recombinant Gal-3	• myocardial fibrosis and its pharmacological inhibition • prevention of remodeling by an inhibitor of Gal-3	[99]
rat	chronic heart failure	intrapericardial infusion of low-dose Gal-3	• increased Gal-3 in hypertrophied hearts • a novel biomarker at the early stages of heart failure	[100]
rat	chronic heart failure	banding of the pulmonary artery	• increase of Gal-3 in ventricles	[101]
rat	ischemia/reperfusion injury	Gal-3 pharmacological inhibition	• Gal-3 blockade improved ischemic injury	[102]
mouse	acute heart failure	viral myocarditis	• time-course analysis of cardiac and serum Gal-3 • an early diagnostic marker for cardiac fibrosis	[13]
mouse	myocardial fibrosis	angiotensin-mediated hypertension in AngII/Cx3cr1-/- mice	• macrophages promoting fibroblast differentiation and collagen production	[96]
mouse	acute myocarditis and chronic fibrosis	coxsackievirus B3-induced myocarditis	• disruption of Gal-3 gene reduced acute myocarditis and chronic fibrosis	[105]
mouse	heart failure	isoproterenol-induced left ventricular dysfunction and fibrosis	• interaction of Gal-3 with aldosterone in promoting macrophage infiltration and cardiac fibrosis	[106]
mouse	pressure-overloaded heart	transverse aortic constriction	• early upregulation of Gal-3 in macrophages • large amounts of Gal-3 in cardiomyocytes at the late stage • Loss of Gal-3 did not affect survival, cardiac fibrosis and hypertrophy	[97]
mouse	acute myocardial infarction	intravenous transplantation of human umbilical cord blood mesenchymal stem cells	• close association of Gal-3 with the ratio of M1 macrophages to M2 macrophages	[108]
mouse	ischemia/reperfusion injury	30 min/24 h in ischemia/ reperfusion model	• contribution of upregulated Gal-3 in cardiac dysfunction • amelioration of myocardial injury by inhibition of Gal-3	[53]
mouse	ischemia/reperfusion injury	wild-type mice and Gal-3 knockout mice	• protective role of Gal-3 on the myocardium following IR injury	[103]
mouse	several mouse models of heart disease	cardiac and plasma Gal-3-level analysis	• multifold increases in cardiac Gal-3 expression • etiology-dependency of increments in circulating Gal-3	[61]
mouse	fibrotic cardiomyopathy	cardiac overexpression of b2-adrenoceptors	• upregulation of cardiac Gal-3 expression • Gal-3 may not be a critical disease mediator of cardiac remodeling	[98]

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
