# Peer review of "Galectin-3: A Potential Prognostic and Diagnostic Marker for Heart Disease and Detection of Early Stage Pathology"

_biomolecules, 2020, doi:10.3390/biom10091277_

Round 1
Reviewer 1 Report
in the review manuscript entitled "Galectin-3 as a diagnostic or prognostic marker for detecting early stage of heart disease" Hara et al. treated recent studies regarding the use of Galectin-3 as a new diagnostic or prognostic biomarker in heart disease. In particular, they discussed clinical and animal model studies about the use of Galectin-3 as a biomarker in several heart disorders. Finally, they performed a short review of other established heart disease biomarkers.
The review treats synthetically but in a comprehensive manner a new important biomarker in heart disease.
There are several minor concerns that need to be addressed.
The review reports a well-described list of clinical and experimental studies but, as reported for NPs, a clear discussion of how Galectin-3 is suggested to be used clinically as an early diagnostic biomarker of cardiac fibrosis in heart failure is absent. What are the putative serum levels that could better discriminate between fibrotic heart failure? What is the clinical algorithm in which Galectin-3 serum levels analysis could be inserted?
Galectin-3 is implicated in many human disorders, not only in heart, fibrosis, or inflammatory diseases (Galectin-3: One Molecule for an Alphabet of Diseases, from A to Z. Int J Mol Sci. 2018;19(2):379.).This could represent an important limitation of the use of serum Galectin-3 for the specific identification of heart disorders. The Authors should better underline this either discussing the contradictory results obtained in clinical studies of Galectin-3 as a possible biomarker of heart disease, either discussing the limits of the use of Galectin-3 as an early biomarker in cardiological disorders.
I suggest the authors insert the synthetic paragraph of the other established cardiovascular biomarkers in the introduction section, for a more comprehensive reading of the review focalized on Galectin-3.
Moreover, the accurate and interesting paragraph on Galectin-3 studies performed in animal models should be treated before the clinical results. This could make the discussion more clear and consequent.
line 41: please read the phrase: "Galectins are a family.......lectins in modulating basic....". Probably some verb has to be added.
line58: Galectin 3 is present also in the cell nucleus where it regulates important molecular processes (eg. splicing).
line 82: Galectin-3 is clinically used in the diagnosis of thyroid tumors. In the discussion of Galectin 3 as a biomarker, this should be included
Reviewer 2 Report
The article by Hara et al reviewed the Galectin-3 as a diagnostic or prognostic marker for detecting early stage of heart disease.
The major concerns are
1) This review is not well focused. For example, this review has vast information about the established cardiovascular biomarkers other than Gal-3 (section 4). This section should be shortened and also the authors should describe how Gal-3 as a marker is better than these markers.
2) Galectin-3 as a diagnostic marker and as a prognostic marker should be discussed in separate subheadings.
3) Information about the mechanism such as how Gal-3 interacts with cell surface receptors and what pathways are activated in the heart is missing. A small paragraph/sub-section in this area will help the readers.
4) The function of cardiac synthesized Gal-3 as well as other cell types synthesized Gal-3 on heart failure is not discussed well.
5) Figure 2: These data are an extension of the previous work published in PLoS One. This figure illustrates the Gal-3 expression in late-stage cardiomyopathy in SG delta knockout mice. It would be appropriate to discuss the relevance of this figure to the theme of this review.
6) Gal-3 should be indicated by arrows in figure 2.
7) Section 8: There are no perspectives and questions that need to be addressed in the conclusion section. This reviewer feels that this is important for a review.
Reviewer 3 Report
In this review article Hara et al summarize the potential of galectin-3 as a marker for heart disease. After introducing the family of galectins (including a relevant diagram) and some general characteristics of galectin-3 the authors further focus on the use of estimations of circulating levels of the latter as diagnostic or prognostic tools in heart disease and failure (table 1) in comparison to other established (NT-proBNP, troponins) or other novel biomarkers (e.g. ST2). The authors point out that galectin-3 elevation depends on macrophage-related inflammation and fibrosis and is not an organ-specific marker. In the next part of the review the authors attempt to summarize results concerning up-regulation of galectin-3 expression and the effect of its inhibition in animal models of cardiac injury and failure. Using a figure and an additional table they conclude that galectin-3 elevation may correlate with the activation status of cardiac macrophages and, furthermore, that it contributes to fibrosis and cardiac dysfunction. They also suggest that galectin-3 levels reflect early inflammatory events and may be used to predict forthcoming myocardial adverse events in heart disease including myocarditis.
Galectin-3 is indeed a recently emerged biomarker of various aspects of heart remodeling and failure both during their initiation and progress, therefore increasingly gaining interest among academia, clinicians and pharmacy, as already depicted in numerous reviews already appeared during the last 10 years. In the meantime, several discrepancies concerning galectin’s-3 usefulness as a marker and target in heart disease have been reported. An update on its potential as such would be therefore particularly important for the cardiovascular oriented audience. The authors correctly included animal data to support their conclusions from the clinical studies and pointed out the correlation of galectin-3 with macrophages. However, there are a lot of gaps in their approach that should be tackled in order to generate a more comprehensive and meaningful review.
In particular:
Whereas the title predisposes the reader to follow the diagnostic-prognostic potential of galectin-3 for early stage of heart disease in general, many examples refer to advanced heart failure. In addition, important aspects of heart disease, such as atrial fibrillation or congenital heart disease, where recent data revealed a role of galectin-3 either as a marker (human studies) or as a contributor (animal models) are missing. Furthermore, the analysis of animal models in the second part included assumptions for galectin’s-3 contribution to heart failure progress, thus its potential as an intervention target. This is also not depicted in the title.
As the primary focus is on the importance of the circulating levels of galectin-3 the characteristics of the unconventional secretion of this lectin by the cells should be introduced, that would more relevant than the many times presented in the literature depiction of the other galectin family members. Furthermore, both human and animal studies identified extracardiac sources of galectin-3 in heart disease, particularly the kidneys (see for instance PMID 29844319). This is rather briefly discussed and should be more elaborately presented. Moreover, the presentation of the additional biomarkers (ST2) is weak, other novel ones (GDF-15) are missing, such a discussion is rather irrelevant in the present review and would be much more meaningful if a mechanistic relation of these biomarkers with galectin-3 existed and could be described.
Both tables are incomplete. There are many more clinical and animal studies conducted on the role of galectin-3 in cardiac disease. A public database estimation on the number of relevant reports would be helpful in the first place. Then the tables could be presented as lists of specific examples. Large metanalyses supporting the importance of circulating galectin-3 in heart failure outcome would be also helpful to include (Imran Am J Cardiol 119, 57, 2017). Furthermore, there are a lot of inconsistencies in the presentation of the cited reports and many important ones are missing. For instance, reference 12 is not included in table 1, despite that report’s conclusion that circulating galectin-3 levels did not correlate neither with its myocardial levels nor fibrosis; reference 78 is included in table 2 and the text to indicate that galectin-3 may be also expressed by cardiomyocytes during the progress of the disease. However, this work also pointed out that in the TAC model galectin-3 appeared not to contribute to fibrosis and dysfunction, emerging a debate (see PMID 29498533). Additional reports suggested that in some animal models galectin-3 may not mediate remodeling and dysfunction (Am J Physiol 314, H1169). This is not presented in the text. In addition, recent reports on the effect of galectin-3 on myocardial infarction and I/R animal models are missing. Additional reports following the action (and source) of galectin-3 in cardiac disease such as atrial fibrosis and fibrillation are also missing.
Moreover, the association of galectin-3 expression and function with the activation status of macrophages is plausible; however the M1-M2 model fades out (Circulation Res 119:414, 2016) and there are many recent reports showing that the macrophage state following a heart injury or insult (for instance myocardial infarction) can not be adequately described by this ex vivo model. The authors should rephrase this part of their discussion.
Finally, the future perspectives and conclusion are mostly plausible assumptions whereas a structured prediction in association with a clearer description of briefly mentioned yet not elaborated novel tools is missing.
Round 2
Reviewer 2 Report
N/A
Reviewer 3 Report
In their revised manuscript, Hara et al considerably improved their review on galectin-3 as a biomarker of heart disease by including more comprehensive representation of other biomarkers such as ST2 and GDF-15 and additional relevant pathologies marked by Galectin-3 (congenital heart disease, atrial fibrillation, CKD). The presented aspects are now better represented in the title. Moreover, they fairly enriched their Tables and discussion on animal models discussing galectin’s-3 potential contribution in cardiac pathophysiology and failure, by including previously missing reports on ischemic models, as well as some publications with negative results. They also normalized their texts on macrophage activation according to the recent evidence on M1/M2 paradigm fading. On the other hand, they did not elaborate on galectin’s-3 secretory mechanisms and still their Perspectives part, although enriched, could be further extended and improved in a reader-friendly comprehensive structure. However, the information on galectin-3 both as biomarker and target of intervention in heart disease is indeed vast and difficult to be included in a medium space report and important aspects of its biology and associated mechanisms are indeed represented in this review.